# Can ChatGPT Counter Vaccine Hesitancy? An Evaluation of ChatGPT’s Responses to Simulated Queries from the General Public

**DOI:** 10.3390/healthcare13111269

**Published:** 2025-05-27

**Authors:** Matthew Chung Yi Koh, Jinghao Nicholas Ngiam, Brenda Mae Alferez Salada, Paul Anantharajah Tambyah, Sophia Archuleta, Jolene Ee Ling Oon

**Affiliations:** 1Division of Infectious Diseases, Department of Medicine, National University Hospital, National University Health System, Singapore 119074, Singapore; matthew.koh@mohh.com.sg (M.C.Y.K.); brenda_mae_alferez@nuhs.edu.sg (B.M.A.S.); mdcpat@nus.edu.sg (P.A.T.); mdcsa@nus.edu.sg (S.A.); jolene_oon@nuhs.edu.sg (J.E.L.O.); 2Yong Loo Lin School of Medicine, National University of Singapore, Singapore 119077, Singapore; 3Infectious Diseases Translational Research Programme, Department of Medicine, Yong Loo Lin School of Medicine, National University of Singapore, Singapore 119077, Singapore

**Keywords:** artificial intelligence, ChatGPT, vaccine hesitancy, vaccinations, education

## Abstract

**Background:** Vaccines have eradicated deadly diseases, yet vaccine hesitancy persists, leading to reduced uptake. Some individuals, mistrustful of healthcare providers, may turn to tools like ChatGPT for information. We evaluated ChatGPT’s responses to common vaccine hesitancy questions. **Methods:** Infectious disease physicians compiled 15 frequently encountered questions relating to vaccine hesitancy, focusing on concerns about efficacy, adverse effects, and cultural or religious issues, and submitted them to ChatGPT on 18 October 2023. Two independent physicians assessed the responses for factual accuracy and alignment with recommendations from the U.S. Centers for Disease Control and Prevention’s Advisory Committee on Immunization Practices (ACIP). **Results:** A representative selection of responses is shown. ChatGPT consistently provided fact-based, reassuring responses. For instance, it accurately addressed the benefits of male HPV vaccination; however, it failed to consider ACIP’s age-specific recommendations and individual sexual health factors. While correcting mRNA vaccine misconceptions, it did not mention the availability of non-mRNA COVID-19 vaccines. It also lacked depth in discussing religious objections, deferring users to faith leaders and providers. **Conclusions**: ChatGPT provides generally accurate information and may be a useful adjunct in addressing vaccine hesitancy. With refinement, it could complement public health efforts to improve vaccine confidence and counter misinformation.

## 1. Introduction

Vaccine hesitancy is a phenomenon of global concern, particularly in the context of the recent coronavirus disease 2019 (COVID-19) pandemic [1]. It covers a broad range of factors that contribute to an individual’s behavioural decision to accept a vaccine [2]. From a public health perspective, this leads to excess morbidity and mortality that could have been prevented with improved uptake of vaccines. One study estimated that non-vaccination against COVID-19 contributed towards greater than 200,000 deaths from COVID-19 in the United States alone [3]. Willingness to vaccinate is influenced by several factors that contribute to an individual’s health beliefs. Easy access to reliable and accurate information, as well as trust in these sources, contributes significantly to an individual’s overall knowledge and attitudes towards vaccines [4].

Individuals who are sceptical of data and information provided by scientific publications on vaccinations may turn to alternate sources for information on the safety and efficacy of vaccines [5]. Several unreliable or overly commercial sources provide inaccurate and sensationalised myths and misinformation on vaccinations, which leads to increased vaccine hesitancy and reduced uptake of important life-saving vaccines [6].

In addition to these sources, individuals may also turn to artificial intelligence (AI) natural language processing chatbots such as ChatGPT for information on vaccines [7]. ChatGPT is widely accessible and provides easy access and responses to a wide range of queries posed by individuals. Its use has been studied in several settings, such as medical education [8,9]. Recently, ChatGPT has been examined in various clinical settings, including pre-travel advice and counselling on safe living practices for solid organ transplant recipients [10,11]. However, its role in providing accurate information and addressing concerns pertaining to vaccine hesitancy remains to be understood and examined. Therefore, from the end-user perspective, we sought to assess ChatGPT’s responses to common vaccine hesitancy concerns, in accordance with factual accuracy and alignment with U.S. CDC Advisory Committee on Immunization Practices (ACIP) guidelines.

## 2. Methods

To do so, we instructed ChatGPT (GPT-3.5) to answer a series of frequently asked queries related to commonly used vaccines that may lead to vaccine hesitancy on 18 October 2023. The date cut-off for ChatGPT’s training data was January 2022 at that time. Fifteen questions were drawn from clinician experience in infectious diseases and vaccine counselling and represented commonly encountered concerns among the general public. We divided these questions into the domains of (i) doubting vaccine efficacy, (ii) concerns about adverse effects, and (iii) cultural concerns pertaining to vaccines. In this exploratory analysis, a qualitative assessment of ChatGPT’s responses was performed independently by two physicians (MCYK and JNN). The responses were checked for factual accuracy and appropriateness against information provided by the ACIP guidelines [12].

## 3. Results

ChatGPT answered all the questions posed on vaccine hesitancy (Table A1). An appraisal of specific questions asked, along with summarized responses given by ChatGPT, is shown in Table 1. Overall, we found that ChatGPT was able to provide highly accurate and up-to-date information when asked about specific vaccines. It was able to address vaccine hesitancy in a balanced manner, providing the pros and cons of vaccination as well as encouraging further discussion with a healthcare professional, to allow for shared decision-making between the individual and their provider.

## 4. Discussion

In the domain of vaccine efficacy, the information provided by ChatGPT was accurate. With regards to measles vaccination, ChatGPT was able to accurately highlight that the benefits of vaccination extend beyond the individual and confer herd immunity in the community. In addition, an individual living in low-prevalence settings for measles may travel to higher endemicity areas in their lifetime, or encounter imported cases of measles, giving an important reason for vaccination at an early age to confer lifelong protection. Indeed, with falling rates of measles vaccination, outbreaks have been reported even in historically low-prevalence areas [13]. When discussing the HPV vaccine for men, ChatGPT was able to explain that the vaccine conferred protection against anal, penile, and oropharyngeal cancers, as well as some protection against non-cancerous conditions like genital warts in addition to cervical cancers. Increasing HPV vaccination in men also confers herd immunity, which helps protect women in the community [14].

However, the responses provided by ChatGPT were generic and not tailored to a specific individual. For example, in trying to convince an individual to get vaccinated against measles, it was important to consider and emphasize specific factors such as the likelihood of an individual to travel to areas of higher endemicity or whether the vaccine itself was mandated by regulatory agencies. There were also minor lapses in the factual accuracy on HPV vaccination. The ACIP recommends routine vaccination in men up to 26 years of age [12], while for those between the ages of 27 and 45, the ACIP recommends shared decision-making between the individual and their healthcare providers. This discussion considers an individual’s risk in the form of sexual health and practices as well as individual preferences. ChatGPT did not consider an individual’s age when making a recommendation for HPV vaccination, but it did direct an individual to discuss further with their healthcare provider for more specific and detailed advice.

In the domain of vaccine adverse effects, the advice provided by ChatGPT was also highly accurate. With regards to the influenza vaccine and egg allergies, the advice provided was up-to-date, highlighting that newer influenza vaccines no longer contain egg proteins, which were previously a concern for patients with severe egg allergies [15]. Rather than being dismissive of the concern, ChatGPT gave the historical context for these initial concerns and then provided reassurance with the latest evidence. ChatGPT was also able to firmly dispel other common myths pertaining to vaccines, such as the permanent alteration to deoxyribonucleic acid (DNA) caused by messenger ribonucleic acid (mRNA) vaccines or the association of the measles, mumps, and rubella (MMR) vaccine with the development of autism in children. In these responses, ChatGPT provided a measured view and explained the mechanism by which vaccines work, with a focus on their safety profiles. When asked about specific vaccine concerns, it took the opportunity to encourage and emphasize the importance of vaccination and prompted the individual to discuss further with a healthcare provider.

ChatGPT was also able to address parental concerns with regards to COVID-19 vaccinations, such as the long-term safety of vaccines. It acknowledged the uncertainty of the vaccine’s long-term side effect profile but reassured the individual that the benefits likely outweigh the risks of vaccination, along with the added reassurance that comes from continued monitoring and safeguards in place. However, although the mRNA vaccines are safe in children, ChatGPT did not provide alternatives such as inactivated COVID-19 vaccines, which may still be an option for parents concerned about the mRNA vaccine [16], or sources of information in the event that the individual or parent is eager to find out more about each specific vaccine.

Finally, when addressing cultural concerns with regards to vaccination, ChatGPT was able to provide balanced views. It demonstrated respect for a person’s religious beliefs while simultaneously encouraging open discussion with religious leaders on this topic. It also provided the scientific facts on the benefits of vaccination for the individual to consider, as these are understandably important concerns to address to improve vaccine uptake in certain religious communities across the world [17]. However, given a more specific context, it may be able to address issues specific to an individual’s religion, rather than keeping its responses vague.

When discussing ethical concerns relating to vaccine manufacturing and development processes, ChatGPT’s answers were sensible and measured. It firmly denied the argument that actual human embryos are used in the testing of mRNA vaccines, without going into the details of which vaccines use foetal cell lines [18]. However, it acknowledged that animal testing does occur in the context of vaccine development, with clear explanations that these studies were conducted with care to protect the welfare of the animals involved and cause the least amount of harm or distress. Animal testing is important to determine the safety and efficacy of vaccines, and contributes to the development of vaccines, which would translate to improving human health. Thus, the responses acknowledged an individual’s concerns but also aimed to provide alternative views to address the individual’s vaccine hesitancy. A further important factor pertaining to COVID-19 is the notion of ‘vaccine fatigue’. Where an individual may have initially accepted vaccines, over time, there may be an increasing inertia towards repeated follow-up vaccination. These may reflect changes in health beliefs, and the increasing misinformation propagated pertaining to vaccines [19]. The measured approach that ChatGPT takes in explaining mRNA vaccines may help to allay some of these fears. However, an important limitation is that this model of ChatGPT also did not discuss the possibility of non-mRNA vaccines, which may be considered for individuals who are sceptical about mRNA vaccines. Additionally, ChatGPT did not cite references for its responses, which may be helpful if individuals were keen to check the veracity of the information by referring to the original source.

Overall, our findings highlight the potential role of AI chatbots like ChatGPT as intermediary tools in public health communication, particularly in the context of addressing vaccine hesitancy in individuals who may distrust traditional sources. There remains a tension between the improved accessibility and personalization of responses from the large language model. For example, GPT-3.5 is easy to access at any time and by anyone, but has trade-offs in lacking specificity to the local context and tends to give generic responses. Additionally, it would be important to consider real-world implementation and evaluate the integration of AI tools into clinical or public health workflows. However, this would still need careful human oversight and regulatory guidance. This is because ChatGPT’s responses still remain limited in terms of the specificity of the advice and response provided. The quality of responses depends on how much information is provided by the individual asking the question. For example, unless details of the specific religious or ethical concerns are given, the responses could only remain generic and not tailored to address an individual’s specific beliefs.

We have previously described the use of ChatGPT as an adjunctive tool for pre-travel advice and counselling. Although it helped to enhance the consultation experience for the traveller, it still could not be used in isolation and was merely an adjunctive tool that required follow-up with further counselling from a human healthcare provider [20]. Furthermore, another important consideration is that the large language model algorithms may also be altered without warning, which may inadvertently affect the performance of these models.

## 5. Limitations

Our study examined a single natural language processing chatbot that was the most widely available. Other chatbots that are designed to answer healthcare-related questions more specifically may be able to provide better responses. As this was a preliminary, hypothesis-generating study, we opted for a qualitative assessment by two independent physicians. We did not employ a formal rubric but checked for factual accuracy and alignment with ACIP recommendations. As such, we were not able to meaningfully provide a numerical percentage of ChatGPT’s responses that adequately met pre-defined standards. Future work could formally assess these responses with a structured rubric and measure inter-rater reliability metrics such as Cohen’s kappa. Also, we did not replicate our study at different timepoints, although with more information over time, artificial intelligence chatbots may also improve on the responses provided. Lastly, the topic of vaccine hesitancy is very broad, so the questions examined could not be fully exhaustive. Also, the questions were selected by healthcare professionals, which lends itself to a degree of bias and may not fully reflect the concerns of the general public. Thus, our findings were meant to be hypothesis-generating, demonstrating that with increasing global vaccine hesitancy and mistrust in science, AI chatbots may help to mediate and correct misinformation on vaccines among the general public.

## 6. Conclusions

ChatGPT’s responses to vaccine hesitancy were mostly accurate and may help to correct misinformation and knowledge in individuals who harbour vaccine misconceptions that lead to a reduced willingness to vaccinate. AI chatbots may be a helpful tool as an adjunct to public health campaigns in eliminating vaccine hesitancy. Future work should include the development of standardized evaluation frameworks for AI tools in health communication. Performance in real-world settings, including evaluating its performance in other languages, would also be an important subject of future study. As newer models are developed, their performance also should be constantly re-evaluated, as these models may be able to overcome the limitations of older models and may be able to provide more personalised and accurate responses.

## Figures and Tables

**Table 1 healthcare-13-01269-t001:** Evaluation of ChatGPT responses to commonly asked questions on vaccine hesitancy.

Question	Summary of ChatGPT Response	Evaluation
*Doubting the efficacy of vaccines*		
I think that big pharmaceutical companies are just trying to profit off us. Their vaccines cannot be trusted. Is that correct?	“The idea that pharmaceutical companies are primarily motivated by profit is a common concern, and there are certainly cases where companies prioritize financial gain over other considerations. However, it’s important to understand that vaccine development and production involve a complex mix of motivations, including public health, scientific advancement, and profitability…”	Overall, ChatGPT highlights important reasons for vaccination, benefiting both the individual and society at large, whilst emphasizing that the vaccine is safe. However, the response provided is generic and not tailored to an individual’s specific situation (for example, how likely is a child to travel to an area with higher risk for measles transmission?). Specific to HPV vaccination, ChatGPT appropriately expounds on the positive benefits of HPV vaccination in men, their sexual partners, and society as a whole in reducing HPV transmission and its associated diseases. However, does not acknowledge the age cut-offs suggested by ACIP for vaccination, or take into account an individual’s sexual health and risk for shared decision-making.
My immune system can handle an infection, and I can also choose to avoid crowds. Why would I still need the COVID-19 vaccine?	“While it’s true that some people may have a robust immune system that can handle infections, and taking precautions like avoiding crowds can reduce your risk of exposure, there are several important reasons to consider getting the COVID-19 vaccine…Protection from severe illness… Reducing transmission…”	
Why do I need to give my child the measles vaccine if it is so rare nowadays?	“Even though measles has become relatively rare… it’s still essential to vaccinate your child… for several important reasons… Herd immunity… resurgence risk… serious health consequences… measles can lead to severe complications… the vaccine is safe and effective… public health… international travel… unvaccinated travelers can bring the virus back to their home country.”	
I am a 28-year-old man. The HPV vaccination prevents cervical cancer. Why should a man like me get the HPV vaccine?	“The HPV vaccine is often associated with preventing cervical cancer in women…important benefits for men as well…protection against…anal, penile and oropharyngeal cancers…preventing transmission…to their sexual partners… herd immunity… reducing… genital warts… provides long-lasting protection… discuss with healthcare provider…”	
Isn’t natural immunity better than vaccination to protect against infections?	“Natural immunity, acquired through exposure to an infectious agent, can provide protection against future infections. However, the effectiveness and safety of natural immunity can vary significantly depending on the specific disease and individual factors…. Risk of severe disease… Consistency of protection…”	
*Concerns about the adverse effects of vaccines*		
Will the MMR vaccine cause autism in my child?	“No, the MMR (measles, mumps, and rubella) vaccine does not cause autism. This myth originated from a flawed and widely discredited study published in 1998 by Andrew Wakefield, which falsely claimed a link between the MMR vaccine and autism. The study was based on a small sample size and had significant methodological issues…”	Advice provided is accurate and up-to-date. With regards to MMR vaccination, ChatGPT unequivocally denies the link with autism, which is the consensus of the scientific community. ChatGPT also highlights that newer influenza vaccines no longer contain egg proteins, which had previously been a concern for patients with severe egg allergies. Rather than being dismissive of the concern, ChatGPT gives the context for these concerns and then provides reassurance. It also consistently prompts the individual to discuss further with a healthcare provider. Pertaining to concerns regarding mRNA vaccines, ChatGPT provides a measured response and explains the mechanism with which mRNA vaccines work, in particular highlighting its safety profile. It does not offer non-mRNA vaccine options. It takes the opportunity to encourage and emphasize the importance of COVID-19 vaccination and prompts the individual to discuss further with a healthcare provider. With regards to vaccination in children, ChatGPT addresses parental concerns with regards to long-term safety of vaccines. While ChatGPT acknowledges the uncertainty of this, it reassures the individual that the benefits likely outweigh the risks of vaccination, along with the added reassurance that comes from continued monitoring and safeguards in place with regards to vaccine safety.
I have an egg allergy. I heard that this means I should not take the flu vaccine. Is that true?	“Having an egg allergy doesn’t necessarily mean you can’t receive the flu vaccine… traditional influenza vaccines were manufactured using chicken eggs… however, many flu vaccines have been reformulated to reduce or eliminate the risk of an allergic reaction in people with egg allergies… consult with a health care provider… egg-free vaccines… FluMist nasal spray…in many cases, individuals with egg allergies can still receive flu vaccines safely.”	
I heard that the COVID-19 vaccine will permanently alter my DNA.	“The idea that COVID-19 vaccines can alter your DNA is a misconception that is not supported by the scientific evidence… mRNA vaccines provide temporary instructions to your cell… vaccine components are broken down and eliminated from your body… mRNA… is fragile and rapidly degrades within your cells… DNA remains unchanged… safe and effective in preventing COVID-19 and its severe consequences.”	
If I let my child get a COVID-19 vaccine, how can we be sure that it will not cause long-term side effects decades later?	“Long-term side effects from vaccines are typically extremely rare… high degree of confidence in the safety of vaccines… Extensive Clinical Trials… Regulatory approval… Ongoing monitoring… historical safety record… impossible to guarantee absence of any long-term side effects with absolute certainty… available evidence and safeguards in place provide a high level of confidence in the safety of COVID-19 vaccines.”	
I heard that the COVID-19 vaccine can cause severe complications like myocarditis. Should I take the COVID-19 vaccine?	“The COVID-19 vaccines have been extensively studied, and they have been found to be safe and effective in preventing COVID-19, a disease that can have serious and even life-threatening complications. While it is true that there have been rare reports of myocarditis (inflammation of the heart muscle) and pericarditis (inflammation of the lining around the heart) occurring after COVID-19 vaccination, it’s important to understand the context of these cases…Rare occurrence… Benefits outweigh risks…”	
Can I get an infection from any kind of vaccine?	“Vaccines are designed to stimulate your immune system to produce an immune response without causing the actual disease. In most cases, vaccines do not contain live, active viruses or bacteria that can cause the disease they are intended to prevent. As a result, you generally cannot get the disease from the vaccine itself…”	
*Cultural concerns pertaining to vaccinations*		
Were the COVID-19 vaccines tested on human embryos?	“The COVID-19 vaccines authorized for use in many countries, such as the Pfizer-BioNTech and Moderna vaccines, as well as the Johnson & Johnson vaccine, were not tested on human embryos. These vaccines were developed and tested using well-established scientific and ethical principles…”	Pertaining to ethical and cultural issues, as stated, the responses that ChatGPT provided were factual and maintained objectivity. Importantly, the responses were also measured and were able to acknowledge an individual’s struggle with the issue. However, because details of the religious concern and specific vaccine details were not given, more detailed discussion on the risks and benefits could not be provided by ChatGPT.
My religion doesn’t allow me to get vaccinated. Is it okay if I choose not to take any vaccines?	“The decision to get vaccinated can be a complex one…key points to consider… public health… religious exemptions… consult with religious leaders… Individual and community impact… informed decision… important to be well-informed about the potential risks and benefits of vaccines and consult with healthcare professionals…”	
It is important for me to start a family. Will the COVID-19 vaccine affect my fertility?	“There is no credible scientific evidence to suggest that the COVID-19 vaccines have a negative impact on fertility. The COVID-19 vaccines authorized for use have undergone rigorous clinical trials and have been closely monitored for safety…”	
Were any animals harmed in the production of vaccines?	“The development and production of vaccines typically involve preclinical testing on animals to ensure their safety and effectiveness before human clinical trials. This is a common practice in pharmaceutical research to better understand how a vaccine or drug may affect living organisms. However, it’s essential to note that there are strict ethical guidelines and regulations in place to minimize harm to animals in research…”	

## Data Availability

Data may be made available on reasonable request from the corresponding author.

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
