# Peer review of "Can ChatGPT Counter Vaccine Hesitancy? An Evaluation of ChatGPT’s Responses to Simulated Queries from the General Public"

_healthcare, 2025, doi:10.3390/healthcare13111269_

Round 1

Reviewer 1 Report

Comments and Suggestions for Authors

The manuscript entitled "How does ChatGPT respond to questions pertaining to vaccine hesitancy from the general public?" addresses a timely and important topic in the intersection of digital health, public communication, and infectious disease. The authors explore the utility of a large language model, ChatGPT, in responding to frequently asked questions regarding vaccine hesitancy. While the manuscript is well-structured and presents a coherent narrative, several methodological and editorial issues warrant further attention prior to publication.

The article demonstrates originality by focusing on the application of AI-generated text in the domain of vaccine communication, a field that remains underexplored despite increasing reliance on digital platforms for public health messaging. The classification of vaccine hesitancy concerns into domains—efficacy, adverse effects, and cultural or ethical considerations—provides a useful framework for analysis. The manuscript benefits from the clinical oversight of infectious disease physicians, lending credibility to the evaluation of ChatGPT’s responses.

Moreover, the inclusion of direct examples of AI-generated content, juxtaposed with commentary from healthcare professionals, enhances the transparency and replicability of the evaluation. The overall tone is balanced, avoiding overstatement of ChatGPT’s capabilities while acknowledging its potential as an adjunct tool in combating misinformation.

Despite its merits, the manuscript exhibits several limitations that should be addressed to improve scientific rigor and interpretability:

The title may be misleading, as it implies that the data were sourced directly from the general public. In reality, the questions were curated by clinicians. It is recommended to revise the title for clarity, for example: “Evaluation of ChatGPT’s Responses to Vaccine Hesitancy: A Simulation of Public Health Queries.”

The abstract summarizes the objectives and findings effectively but lacks methodological specificity. It would benefit from including the date of interaction with ChatGPT, the total number of questions evaluated, and the assessment criteria used.

The introduction is generally well-written and contextualizes the issue of vaccine hesitancy. However, it lacks a clearly articulated research objective. A more explicit research aim, such as "to assess the factual accuracy and guideline adherence of ChatGPT’s responses to vaccine-related concerns," would strengthen the section.

This section is underdeveloped and insufficiently detailed for replication. The authors do not specify which version of ChatGPT was used, nor the date of its last training data update. Furthermore, the process of question selection lacks transparency. The absence of a structured evaluation tool—such as a rubric or scoring system for factual accuracy, clinical alignment, or completeness—limits the reliability of the physician assessments.

Additionally, no inter-rater reliability metrics (e.g., Cohen’s kappa) are reported, which raises concerns about the subjectivity of the reviewers' judgments.

The findings are clearly presented, and the excerpts are illustrative. However, the analysis is predominantly qualitative. There is no summary of how many responses were fully aligned with ACIP guidelines, partially aligned, or omitted key information. Including basic quantitative data (e.g., percentage of responses meeting predefined standards) would substantially improve the robustness of the findings.

Moreover, the results would be enhanced by the inclusion of a summary table categorizing responses by domain and evaluative outcome.

The discussion is well-developed and includes insightful reflections on the potential utility and limitations of ChatGPT in addressing public health misinformation. Nevertheless, it reiterates several points from the results section without deepening the interpretive or theoretical insights.

The authors should consider discussing ChatGPT’s potential role within a broader health communication framework. For instance, could it serve as a triage tool before healthcare consultation? What are the risks of over-reliance on AI-generated content?

The omission of non-mRNA vaccine options in responses, and the lack of reference citations in the chatbot’s answers, should be further critiqued as limitations in user-centered care.

While the limitations are acknowledged, they are discussed superficially. The manuscript should emphasize the non-exhaustive nature of the questions, the static nature of the AI model’s responses at a single timepoint, and the absence of user-specific tailoring. Additionally, the potential bias introduced by clinician-generated prompts warrants discussion.

The conclusions are appropriate but general. The authors may wish to strengthen their final remarks by recommending the development of standardized evaluation frameworks for AI tools in health communication. Future research directions, including multilingual evaluation and engagement with real-world users, should be outlined.

Comments on the Quality of English Language

Overall, the manuscript demonstrates competent use of English, with a formal tone appropriate for scientific publication. The writing is generally clear and understandable, and the authors employ relevant terminology and domain-specific vocabulary correctly. However, there are several areas where linguistic and stylistic improvements would enhance the manuscript’s clarity, consistency, and professionalism.

Author Response

Manuscript ID: healthcare-3662250

How does ChatGPT respond to questions pertaining to vaccine hesitancy from the general public?

Healthcare

We thank the Editor for allowing us the opportunity to revise our manuscript and the Reviewers for the important and constructive comments. We have amended our paper in order to address the points raised by the Reviewer and Editorial Office.

In the sections below, each of the points raised is identified and addressed with changes in the revised manuscript.

The manuscript entitled "How does ChatGPT respond to questions pertaining to vaccine hesitancy from the general public?" addresses a timely and important topic in the intersection of digital health, public communication, and infectious disease. The authors explore the utility of a large language model, ChatGPT, in responding to frequently asked questions regarding vaccine hesitancy. While the manuscript is well-structured and presents a coherent narrative, several methodological and editorial issues warrant further attention prior to publication.

The article demonstrates originality by focusing on the application of AI-generated text in the domain of vaccine communication, a field that remains underexplored despite increasing reliance on digital platforms for public health messaging. The classification of vaccine hesitancy concerns into domains—efficacy, adverse effects, and cultural or ethical considerations—provides a useful framework for analysis. The manuscript benefits from the clinical oversight of infectious disease physicians, lending credibility to the evaluation of ChatGPT’s responses.

Moreover, the inclusion of direct examples of AI-generated content, juxtaposed with commentary from healthcare professionals, enhances the transparency and replicability of the evaluation. The overall tone is balanced, avoiding overstatement of ChatGPT’s capabilities while acknowledging its potential as an adjunct tool in combating misinformation.

We thank the Reviewer for the overall positive feedback on our manuscript. We are grateful for the opportunity to revise and improve our manuscript.

Despite its merits, the manuscript exhibits several limitations that should be addressed to improve scientific rigor and interpretability:

The title may be misleading, as it implies that the data were sourced directly from the general public. In reality, the questions were curated by clinicians. It is recommended to revise the title for clarity, for example: “Evaluation of ChatGPT’s Responses to Vaccine Hesitancy: A Simulation of Public Health Queries.”

We thank the Reviewer for this excellent point. We have updated our manuscript title for clarity. It now reads as suggested, “Can ChatGPT counter vaccine hesitancy? An evaluation of ChatGPT’s responses to simulated queries from the general public” (Page 1)

The abstract summarizes the objectives and findings effectively but lacks methodological specificity. It would benefit from including the date of interaction with ChatGPT, the total number of questions evaluated, and the assessment criteria used.

We have improved on the abstract to enhance the methodology (Page 2).

The introduction is generally well-written and contextualizes the issue of vaccine hesitancy. However, it lacks a clearly articulated research objective. A more explicit research aim, such as "to assess the factual accuracy and guideline adherence of ChatGPT’s responses to vaccine-related concerns," would strengthen the section.

We thank the Reviewer for this comment. We have enhanced our introduction for improved clarity as suggested (Page 3).

This section is underdeveloped and insufficiently detailed for replication. The authors do not specify which version of ChatGPT was used, nor the date of its last training data update. Furthermore, the process of question selection lacks transparency. The absence of a structured evaluation tool—such as a rubric or scoring system for factual accuracy, clinical alignment, or completeness—limits the reliability of the physician assessments.

Additionally, no inter-rater reliability metrics (e.g., Cohen’s kappa) are reported, which raises concerns about the subjectivity of the reviewers' judgments.

We thank the reviewer for these valuable comments. In response, we have clarified the following methodological details (Pages 3-4) in the revised manuscript:

We now explicitly state that we used ChatGPT (GPT-3.5) on 18 October 2023, and the cut-off date for its training data at the time was January of 2022.

We have elaborated on the process by which the 15 questions were developed—these were drawn from clinician experience in infectious diseases and vaccine counseling, and represent commonly encountered concerns across three domains: vaccine efficacy, safety, and cultural/religious issues. This rationale has been added to the Methods section for greater transparency.

As this was a preliminary, hypothesis-generating study, we opted for a qualitative assessment by two independent physicians with expertise in immunization practices. While we did not employ a formal rubric, we have now clarified that responses were evaluated for factual accuracy and alignment with ACIP recommendations. We acknowledge that the absence of a structured scoring rubric and inter-rater reliability metrics such as Cohen’s kappa is a limitation, and have added this to the Limitations section (Page 9).

The findings are clearly presented, and the excerpts are illustrative. However, the analysis is predominantly qualitative. There is no summary of how many responses were fully aligned with ACIP guidelines, partially aligned, or omitted key information. Including basic quantitative data (e.g., percentage of responses meeting predefined standards) would substantially improve the robustness of the findings.

Moreover, the results would be enhanced by the inclusion of a summary table categorizing responses by domain and evaluative outcome.

We thank the Reviewer for this comment. This was an exploratory analysis, and the list of questions examined were not exhausted. We also did not have structured rubrics to define if responses had met pre-defined standards. As such we were unable to provide the percentage of responses meeting the standards. Although we were not able to provide the percentage of questions meeting the standard for each domain, we were able to summarize qualitatively the key strengths and weaknesses of ChatGPT’s responses in each domain. As an exploratory study, we believe this would be helpful in identifying the potential benefits and pitfalls of using ChatGPT to address vaccine hesitancy.

We have added this to our limitations (Page 9).

The discussion is well-developed and includes insightful reflections on the potential utility and limitations of ChatGPT in addressing public health misinformation. Nevertheless, it reiterates several points from the results section without deepening the interpretive or theoretical insights.

We appreciate the reviewer’s positive feedback on the discussion and the suggestion to enrich its interpretive depth. In response, we have revised the discussion (Page 8) to reduce repetition of descriptive results and instead highlight broader implications, including:

  • The potential role of AI chatbots like ChatGPT as intermediary tools in public health communication, particularly for vaccine-hesitant individuals who may distrust traditional sources.

  • The tension between accessibility and personalization in large language models, and how this trade-off might affect trust and uptake of vaccine information.

  • Considerations around integrating AI tools into clinical or public health workflows, including the need for human oversight and regulatory guidance to mitigate misinformation risks.

The authors should consider discussing ChatGPT’s potential role within a broader health communication framework. For instance, could it serve as a triage tool before healthcare consultation? What are the risks of over-reliance on AI-generated content?

We thank the Reviewer for this comment. Indeed, there are important considerations for the use of ChatGPT in a clinical context. It may be used an adjunctive tool for patient counselling, and we have described real-world experience with using this for pre-travel advice. However, some limitations remain, and it still requires the detailed counselling of a human healthcare provider. Furthermore, the AI/LLM algorithms may also be altered without warning, which may affect the performance of these models. This is expanded on in our discussion (Page 8).

The omission of non-mRNA vaccine options in responses, and the lack of reference citations in the chatbot’s answers, should be further critiqued as limitations in user-centered care.

We thank the Reviewer for these points. We have added these important points in our discussion (Page 8).

While the limitations are acknowledged, they are discussed superficially. The manuscript should emphasize the non-exhaustive nature of the questions, the static nature of the AI model’s responses at a single timepoint, and the absence of user-specific tailoring. Additionally, the potential bias introduced by clinician-generated prompts warrants discussion.

We thank the Reviewer for pointing this out. We have significantly enhanced our limitations section to reflect these important points (Page 9).

The conclusions are appropriate but general. The authors may wish to strengthen their final remarks by recommending the development of standardized evaluation frameworks for AI tools in health communication. Future research directions, including multilingual evaluation and engagement with real-world users, should be outlined.

We thank the Reviewer for this comment. We have improved on our conclusion to provide more detail and clarity (Page 9-10).

Overall, the manuscript demonstrates competent use of English, with a formal tone appropriate for scientific publication. The writing is generally clear and understandable, and the authors employ relevant terminology and domain-specific vocabulary correctly. However, there are several areas where linguistic and stylistic improvements would enhance the manuscript’s clarity, consistency, and professionalism.

We thank the Reviewer for this feedback. We have improved on the English and corrected minor grammatical errors in our manuscript.

Once again, we thank the Reviewer the Editor for the opportunity to revise our manuscript and improve on it. We hope that it is now suitable for publication in the Journal.

Best Regards,

Dr Matthew Koh and Dr Nicholas Ngiam

Reviewer 2 Report

Comments and Suggestions for Authors
  1. It would be a good idea to have a brief discussion of how the AI could improve its responses, especially for more sensitive and complicated cultural issues such as certain religious objections to vaccination (Lines 125-130).
  2. The study primarily focuses on ChatGPT's performance in addressing vaccine-related concerns (Lines 58-59). It may be useful to compare its effectiveness with other AI models specifically tailored for healthcare-related queries, which could offer deeper insights into its strengths and limitations (Lines 149-150).
  3. Although the study has mentioned concerns about mRNA vaccines (lines 109-110), it does not state the availability of alternative options for mRNA vaccine hesitaters including inactivated COVID-19 vaccines. This addition could generate a more balanced view and give comfort for the alternative's seekers (line121-122).

Author Response

Manuscript ID: healthcare-3662250

How does ChatGPT respond to questions pertaining to vaccine hesitancy from the general public?

Healthcare

We thank the Editor for allowing us the opportunity to revise our manuscript and the Reviewers for the important and constructive comments. We have amended our paper in order to address the points raised by the Reviewer and Editorial Office.

In the sections below, each of the points raised is identified and addressed with changes in the revised manuscript.

It would be a good idea to have a brief discussion of how the AI could improve its responses, especially for more sensitive and complicated cultural issues such as certain religious objections to vaccination (Lines 125-130).

We thank the Reviewer for this comment. We believe the AI chatbot could improve by not only providing the scientific facts on the benefits of vaccination for the individual to consider, but also address specific concerns raised by the individual’s religious beliefs.  If the AI chatbot had been given more specific context (e.g. the specific religious belief of the individual), it may be able to address issues specific to an individual’s religion, rather than keeping its responses vague. This has been added to the discussion (Page 6-7).

The study primarily focuses on ChatGPT's performance in addressing vaccine-related concerns (Lines 58-59). It may be useful to compare its effectiveness with other AI models specifically tailored for healthcare-related queries, which could offer deeper insights into its strengths and limitations (Lines 149-150).

Indeed, we thank the Reviewer for this comment. We chose GPT-3.5 as it was the free-to-access model at the time of writing. It was the most publicly available and the model that would be most likely used by the general public to address their queries or concerns. Future work evaluating the model against AI chatbots that are specific to healthcare-related queries would be an interesting topic for future study. This has been added to our limitations as well (Page 9).

Although the study has mentioned concerns about mRNA vaccines (lines 109-110), it does not state the availability of alternative options for mRNA vaccine hesitaters including inactivated COVID-19 vaccines. This addition could generate a more balanced view and give comfort for the alternative's seekers (line121-122).

The Reviewer raises a very important point. We have expanded upon this in our discussion (Page 7). It is an important limitation of GPT-3.5’s response, where it did not mention the availability of alternative options such as non-mRNA vaccines.

 Find the additional comments:

  • The topic is novel and fills a gap in understanding how AI tools like ChatGPT may influence vaccine confidence. Consider mentioning more explicitly how this study differs from previous chatbot or AI-related health studies.

Most studies try to evaluate AI as a medical tool to assist physicians. Here, we consider that ChatGPT is already widely used by the general public. Instead of looking at it from the physician’s point of view, we consider the end-user. We expect that patients may already be asking ChatGPT their concerns pertaining to vaccine hesitancy. As such, here, we appraise ChatGPT’s responses, which may help physicians understand the strengths and limitations of this tool that patients may use, in other to better provide counselling for patients. We have expanded on this in our introduction (Page 3).

  • The paper provides valuable observations on the practical use of AI in public health communication. However, it would be helpful to expand slightly on how these findings might guide public health agencies or developers in refining chatbot responses.

We have expanded on these points in our discussion. There is potential for this to be used as an adjunctive tool. We describe a use-case where we have implemented ChatGPT as an adjunctive tool in a clinic for pre-travel advice and counselling. However, several important limitations and pitfalls remain, and it cannot fully replace the human healthcare provider. These points are expanded on with greater clarity in our discussion (Page 8).

  • How were the 15 questions selected (e.g., found in literature search, under-represented topic, based on expert opinion, other)?

We have clarified in our methods that these questions were selected by expert opinion. Infectious Diseases physicians selected these questions as what is deemed to be most representative questions that patients may ask pertaining to vaccine hesitancy. In other words, these are the questions that the physicians find that they would commonly have to respond to in clinical settings. We have clarified this in our methods (Page 3-4)

  • Clarify how the two physicians evaluated—whether it was a rating scale or assessment criterion that was used.

The evaluation was done qualitatively. The responses were checked for appropriateness as well as factual accuracy, by comparing them against guidelines from the ACIP. Future work would include the development of structure rubrics that would provide more consistent and objective grading, as well as measurement of inter-rater variability. This is added to our methods and limitations (Page 3-4, 9).

  • A short statement about whether the doctors’ read-outs were scored independently or negotiated out would be helpful information.

The questions were independently graded by each physician. This is added to our methods (Page 4).

The conclusions are consistent with the findings in general. To strengthen them:

  • Clarify how the constraints (e.g., the absence of personalisation) could be addressed in ChatGPT in future iterations and in other models like it.
  • Briefly explain how these observations might be used to inform AI training or for designing chatbots.
  • The references are relevant and recent. Still, consider adding 1–2 citations that discuss AI usability or limitations in patient communication to give further context to ChatGPT’s role and its boundaries in healthcare.

We thank the Reviewer for this comment. We have added a more recent citation on a use case of implementation of ChatGPT as an adjunctive tool in a pre-travel clinic setting (ref 20). We have also described now newer versions of LLMs can help address some of these previously described limitations (Page 9-10).

Once again, we thank the Reviewer the Editor for the opportunity to revise our manuscript and improve on it. We hope that it is now suitable for publication in the Journal.

Best Regards,

Dr Matthew Koh and Dr Nicholas Ngiam